# Anatomical Considerations for Endovascular Intervention for Extracranial Carotid Disease: A Review of the Literature and Recommended Guidelines

**DOI:** 10.3390/jcm9113460

**Published:** 2020-10-27

**Authors:** Tyler Scullen, Mansour Mathkour, Christopher Carr, Arthur Wang, Peter S. Amenta, John D. Nerva, Aaron S. Dumont

**Affiliations:** 1Department of Neurological Surgery, Tulane Medical Center, New Orleans, LA 70130, USA; tylerscullen@gmail.com (T.S.); mansour.mathkour@gmail.com (M.M.); christopher.carr1984@gmail.com (C.C.); awang15@tulane.edu (A.W.); pamenta@tulane.edu (P.S.A.); jnerva@tulane.edu (J.D.N.); 2Department of Neurological Surgery, Ochsner Medical Center, Jefferson, LA 70121, USA

**Keywords:** Carotid Artery Disease, Endovascular Procedures, Mechanical Thrombectomy, Carotid Stent

## Abstract

Patient selection for endovascular intervention in extracranial carotid disease is centered on vascular anatomy. We review anatomical considerations for non-traumatic disease and offer guidelines in patient selection and management. We conducted a systematic literature review without meta-analysis for studies involving anatomical considerations in extracranial carotid intervention for non-traumatic disease. Anatomical considerations discussed included aortic arch variants, degree of vessel stenosis, angulation, tortuosity, and anomalous origins, and atheromatous plaque morphology, composition, and location. Available literature suggests that anatomical risks of morbidity are largely secondary to increased procedural times and difficulties in intervention system delivery. We recommend the prioritization of endovascular techniques on an individual basis in cases where accessible systems and surgeon familiarity provide an acceptable likelihood of rapid access and device deployment.

## 1. Introduction

Rapid advances in endovascular surgery have led to paradigm shifts in the management of cerebrovascular disease [1]. Increasing interventions of the anterior intracranial circulation for vascular malformation, aneurysm, and acute ischemic stroke (AIS) have prompted numerous studies centering on extracranial access and local anatomy of the internal (ICA), external (ECA), and common carotid arteries (CCA) [2,3,4,5,6]. Likewise, endovascular intervention in extracranial carotid disease has risen in popularity, with active investigation becoming increasingly centered on vascular anatomy in patient selection [7].

As technology and techniques continue to progress, many of the anatomical constraints historically deemed hostile to endovascular intervention have become manageable with advanced methods [1,2,3,4,5,6,7]. In this report we review anatomical considerations in the context of extracranial endovascular carotid intervention for non-traumatic disease and offer guidelines in patient selection and management.

## 2. Results

We conducted a systematic literature review without meta-analysis per Preferred Reporting Items for Systematic Reviews and Meta-Analyses (PRISMA) guidelines [8]. Nineteen studies were identified that discussed perioperative risk in the context of relevant anatomy [9,10,11,12,13,14,15,16,17,18,19,20,21,22,23,24,25,26,27] (Table 1) in addition to fourteen studies [28,29,30,31,32,33,34,35,36,37,38,39,40,41] (Table 2) and fifteen case reports/series [42,43,44,45,46,47,48,49,50,51,52,53,54,55,56] (Table 3). 

Five studies discussed changes in anatomy brought on by intervention [57,58,59,60,61] (Table 4).

All studies regarded intervention by either carotid artery stenting (CAS) and/or balloon percutaneous transluminal angioplasty (PTA) with or without the use of embolic protection devices (EPD), endovascular mechanical thrombectomy (EVT) for large vessel occlusion (LVO), or combinations of the above. Reports presented retrospectively analyzed data from retrospective or prospective databases or as post hoc analyses of prior prospective clinical trials. Anatomical considerations discussed included aortic arch elongation or great vessel origin variants (47%), degree of ipsilateral or contralateral CCA or ICA stenosis (26%), CCA/ICA angulation, tortuosity, or anomalous vessel origins (34%), atheromatous plaque morphology, intimal involvement, composition, and location (57%), or altered hemodynamics (8%). Studies with univariate or multivariate analysis most often examined endpoints of cerebrovascular accident defined by ipsilateral AIS or transient ischemic attack (TIA) (83%), myocardial infarction (MI) (41%), mortality (34%), or the presence of new perioperative restricting parenchymal lesions on magnetic resonance imaging diffusion weighted images (MRI DWI) (14%). Outcomes were most often measured in the short-term follow up period of thirty days or less (76%) with the minority providing data from follow-up one year or more (21%).

### Studies with Multivariate Analysis

Multivariate analysis was provided in nine series [14,16,17,18,19,20,22,26,27]. Associated immediate outcomes included the development of post-operative hemodynamic depression (HD), independently predicted by degree of change in pre and post-CAS ICA angulation while plaque morphology and ICA ostium involvement were not significantly associated [26], and the detection of macroscopic intraluminal debris following stent placement, which was predicted by the presence of a heterogeneous echolucent plaque on enhanced carotid ultrasound (CEUS) [27]. Short-term outcomes included the presence of new restriction on DWI at 24 h reported in three studies to be predicted by severe ICA tortuosity and the presence of aortic arch or CCA plaques [14,18,19]. Severe CCA tortuosity or atheroma, ECA atheroma, contralateral ICA occlusion, type III arch, ICA angulation, and plaque length and morphology were all found to be not significantly associated [14,18,19]. Three studies reported on the presence of symptomatic AIS at one month, with predictors including heavy concentric calcification and catheter manipulation time, while the Delphi anatomic score and the presence of a type III arch were not associated with outcome [17,20,22]. Restoration of healthy ICA peak systolic velocities (vICA) less than 120 cm/s with a concurrent vICA:vCCA ratio less than 1.4 at medium-term follow up 30 months was independently predicted by decreased plaque calcification, concentric calcification, and perilesional soft plaque [16].

## 3. Discussion

CAS has risen as an alternative to carotid endartectomy (CEA) in the treatment of symptomatic extracranial carotid stenosis in appropriately selected patients [1,62,63,64,65,66,67,68,69]. Recent trial data comparing the two was found to be equivocal for short and long-term primary endpoints, and continued analysis in pooled studies reported increased risk of AIS in the first month following CAS [70,71,72]. As such, early guidelines recommended CAS to be reserved as an alternative to CEA in anatomically favorable scenarios [1,9,10,11,12,13,14,15,16,17,18,19,20,21,22,23,24,25,26,27]. Anatomic factors often dictate treatment outside of CAS, such as during emergent EVT and PTA in the context of AIS secondary to LVO or intima dissection [73]. Recent years have seen significant progress in device design and endovascular technique, and the rate of progress in effective treatment has outpaced formal indications and guidelines [74,75], making the latter increasingly inapplicable. Likewise, the recommendations placed regarding the indications and contraindications of CAS and other carotid interventions are based on well-designed trials presenting data from methods and technology that has since been replaced or improved [51,52,53,54,55].

### 3.1. Aortic Arch Angioarchitecture

Consequent of flow dependent development, the adult thoracic aortic arch is highly variable [76]. Although rare configurations such as double arches, right-sided arches, and aberrant vessel origins may occur, most variations occur at the origins of the three great vessels; the brachiocephalic, the left CCA, and the left subclavian artery [76]. The majority of individuals have the great vessels origins lying in the same coronal plane (type I arch), with alternative patterns including elongation variants (type II and III arches), ICA and CCA angulations, and branch variants [76]. Shared origins of the left CCA with, or a left CCA origin off of the brachiocephalic artery occurs in roughly one quarter of patients [76]. Colloquially referred to as a ‘bovine arch’ (though a true bovine arch shows a common trunk for all three vessels and is far more rare), a steeper angle exists relative to the direction of flow of the left CCA as it relates to catheters attempting to access from across the distal transverse segment of the thoracic aorta when transfemoral access (TFA) via the common femoral artery (CFA) is used [76]. The resulting angulated path of the catheter system results in a significant coronal torque on the distal end, increasing the odds and magnitude of contact with the vessel wall, leading to potential intimal injury, inability to advance, and herniation of the advancing catheter into the arch resulting in loss of access [76,77]. Similar force alternations are observed in cases where the CCA or ICA has a sharp angulation at its origin, particularly when the ratio of CCA:ICA angulation is greater than 60 degrees [12].

Relief of forces generated by an unfavorable pivot point can be achieved via a number of dedicated guided catheters and sheaths by adding flexibility and support, although many of devices used in procedures such as CAS have a rigid composition with limited tolerance to curving around an acute angle [76,77]. Additionally, the composite forces added increase the difficulty of controlling minute and precise movements at the region of interest and can lead to potential errors in CAS or EPD delivery [76,77]. While transradial (TRA) and transbrachial access (TBA) has been shown to limit complication rates in regards to intracranial stent delivery [6] as well as in case series regarding extracranial intervention [8,32,34,38,41], the same steep angles and pivots points can still be encountered in unfavorable variants relative to any access site.

Available literature shows arch and great vessel architecture heavily studied for associated with short- and long-term morbidity, particularly AIS. Analytic conclusions are markedly variable, as while over 50% of relevant studies conclude arch/great vessel anatomy to significantly effect outcomes, there is no consensus regarding any one specific variant [9,10,11,12,13,14,15,16,17,19,20,23,26]. Retrospective studies of data extracted from the Endarterectomy versus Stenting in Patients with Symptomatic Severe Carotid Stenosis trial (EVA-3S) implicate the presence of type III arch variants [15], colloquial bovine arch [12], and angulation of the ICA or CCA [11,15], with an increased risk of AIS or TIA at 30 day time points on univariate analysis. Conversely, other reports have failed to show association of arch type or ICA/CCA angulation with peri-operative morbidity [18], and reviewed multivariate studies uniformly reported no association of any one anatomic characteristic with clinical or radiographic complications at one month [14,17,18,20]. One such study, conducted across 282 patients who underwent CAS for asymptomatic or symptomatic carotid stenosis, found that while arch anatomy and angulation was not found to be associated with morbidity at 30 days, Type II and III elongation variants correlated with increased catheter manipulation time (CMT) reported an independent predictor of AIS and MI [17]. 

Speculatively, difficult arch anatomy likely increases the technical difficulty of a given procedure and thus leads to longer operative time and CMT. Increased CMT in turn increases the odds of vascular injury as increased movements of catheter systems evoke greater incidences of intima contact and risk intimal injury, dissection, or atheroemboli. Furthermore, a recent retrospective across 224 patients reported that patients with Type III arch variants had no significant changes in AIS, MI, or mortality as compared to a pooled population of Type I and II variants, but had a higher incidence of contrast- induced nephropathy (CIN) in the immediate post-operative period [23]. Given the direct relationship of CIN to iodinated contrast volume administered, which is in turn related to the number of angiographic injections and length of procedure, the finding seems natural.

Outcome discrepancies in difficult arch anatomy thus may relate to difficult analysis of heterogeneous and likely synergistic individual features along with surgical experience. In combination, various anatomic challenges can increase CMT and thus the risk of AIS, although studies are needed for clarification.

### 3.2. Patient Age

Reviewed studies inconsistently implicate ischemic events to be associated with arch anatomy and branch angulation, but also note the significance of advancing age as a predictor [12]. Further pooling various separate anatomic features into informal risk categories demonstrate significant association of ‘high risk’ anatomy with age over 80 years [12]. Although there appears to be increased risk of AIS in elderly patients undergoing CAS [78], univariate analysis of complication rates between anatomic risk categories while stratifying elderly (over 80 years of age) and younger patients undergoing CAS [79], making it difficult to define a causal relationship regarding arch variations. Illustratively, univariate studies regarding populations separated by age found no association with perioperative morbidity [79,80]. It is likely that while age increases risk of adverse anatomy, it is a confounding variable and the anatomy itself is a predictor to potential complications [79,80,81,82,83].

### 3.3. Internal Carotid Artery Tortuosity

Similar to complex aortic arch anatomy, severe ICA or CCA tortuosity (defined as the presence of kinks with acute angulation less than 90 degrees, loops with a ‘C’ or ‘S’ shaped curvature and two or more areas of angulation less than 90 degrees, or coils with full 360 degree turns) demands increased CMT through often-fragile vasculature [84,85]. Furthermore, turbulent hemodynamics create lumen that is difficult to navigate [84,85,86,87], and excessive curve necessitate flexible stents to prevent kinking, often requiring an open cell design, which in itself can create further challenges in the context of complex plaque morphology [79,88]. Severe tortuosity also increases the difficulty of EPD employment, which is reported with varied effectiveness in reducing atheroembolic events [79]. CCA and ICA tortuosity has been reported to correlate with MI, mortality, AIS and TIA at 30 days [12,15] and new DWI restriction at 24 h [14]. The presence of severe ICA, although not CCA tortuosity, has been reported as an independent predictor of the latter on multivariate analysis [14].

### 3.4. Internal and Common Carotid Artery Degrees of Stenosis

Conceptually, the extent of ICA or CCA stenosis narrows working space and risks potential intima damage and atheroemboli, while impeding safe access and treatment [13,79]. Flow through intracranial anterior and posterior circulations has been shown to elevate abruptly after CAS deployment in severe stenosis, and slowly receded to baseline over three months [61]. Sudden corrections of stenotic lumens have been consistently reported to elevate flow through the diseased vessel and may tax the autoregulatory mechanisms of cerebral microvasculature, leading to increased risk of hyperperfusion injury on univariate analysis [24].

Chronic high-grade ICA stenosis or occlusion triggers adaptive vascular remodeling and the development of collateral pathways to provide adequate flow at baseline [24]. Symptomatic lesions are likely secondary to global hemodynamic depression or contralateral occlusion, and the benefit in treating such lesions is questionable and must be weighted individually. Accentuated risks include hyperperfusion injury and iatrogenic intimal injury, and heavy consideration in treatment must be paid to movements when navigating the lesioned area [24].

Literature is limited to case series, reporting relatively low rates of symptomatic AIS (1%, 4/412) on long-term follow up following CAS in carefully selected patients [33,36,40]. Patients in these cases were first treated with balloon PTA, allowing safe passage and deployment of an EPD prior to CAS [33,36,40]. PTA is carried out before EPD deployment, as the severity of stenosis physically constricts and prevents the EPD delivery catheter from crossing the balloon nose cone. The presence of DWI restriction without correlated clinical AIS or TIA suggests that patients in these cases do experience periprocedural atheroembolic showers, although the majority of such events may not be clinically significant [36]. Despite this finding, the association of showering atheroemboli as evident by DWI restriction suggests a failure in distal embolic protection during CAS. Chronic high-grade stenosis in symptomatic patients should be heavily considered against alternative treatment options including CEA and medical management due to uncertain safety profile in revascularization.

### 3.5. Plaque Morphology and Location

Multiple studies have analyzed the impact of plaque location, composition and echogenicity, and morphology on progression to intimal damage and development of intraluminal atherothrombi [89,90]. Concentric and calcified plaques of the ICA pose intraluminal access and treatment barriers and have been reported to be associated with a negative outcome at 30 days [15,16,19,20]. Calcified plaques coincide with rigid and non-compliant vasculature, decreasing the effectiveness of PTA prior to CAS and physically hindering catheter system deployment [16,19]. The resulting failed vessel dilation with stent material further decreases vessel diameter and flow and increases the likelihood of thromboembolic events [20]. Accordingly, heavy concentric plaque calcification at the site of treatment has been reported an independent predictor of AIS at one month, and the severity and thickness of calcification inversely associated with achieving optimal vICA flow and vICA:vCCA ratios at 30 month follow up on multivariate analyses [16,20].

Plaques within the aortic arch or CCA may be disrupted during catheter passage, which can shear fragile intima and lead to atheroembolism with minimal contact [19]. While the presence of arch plaques has correlated with new restriction on DWI at 24 h on multivariate analysis [19], reviewed multivariate data on CCA lesions is conflicting [18,19]. Likely such plaques increase procedural complexity, and observed risk is secondary to longer CMT, as had been implicated in other anatomical features [17].

Additional challenges exist in so-called vulnerable plaque morphologies [25,27,42,52]. The Stary classification divides plaque grading into six categories, with type V (fibroatheroma) and VI (complicated) lesions considered vulnerable [91,92,93,94]. Histopathologically vulnerable features have included the presence of intraplaque hemorrhage, a necrotic lipid-rich core, and ruptured fibrous caps [93], all of which are correlated with increased T1 and T2 weight MRI signal intensity ratios (SIR) on multivariate analysis [93].

The use of vessel wall imaging via high-resolution T1 inversion-recovery prepared sampling perfection with application-optimized contrast found SIR and the presence of surface irregularities to be significantly associated on multivariate studies with confirmed atheroembolic events [95]. Plaque enhancement was not found to be predictive [96]. Advanced sequences using T1 MRI to compare plaque signal intensity to the sternocleidomastoid muscle, dubbed plaque:muscle ratio (PMR) index panels, found the atheromatous predominant subdivision of type V (Va1) to correlate with a PMR greater than 1.51 in lesions resected during CEA [94]. These lesions in turn were found to be independent predictors of new DWI lesions on logistic regression [93] and correlate with decreased restoration of flow at 30 months [16] in patients undergoing CAS. Colloquially known as ‘soft plaques’, Va1 plaques are heterogeneously echolucent on ultrasound, as with low signal from lipid cores and high signal from fibrous layers as they progress towards thrombus formation, and are associated with a 12-fold increased risk of periprocedural macroscopic debris post-CAS with EPD [27].

Such lesions may rupture and shower atheromatous debris [27] with minimal stimulation, induce formation of occlusive intraluminal mobile flaps [52] or free floating thrombi (FFT) [42], and trigger in-stent stenosis via thrombosis or intimal hyperplasia following CAS [25]. Furthermore, flexible open-cell stents suitable to navigating tortuous anatomy have been reported to risk a scissoring effect following stent expansion and collapse in vulnerable plaques, in which individual threads of the stent shear embolic material [27]. Such findings have been reported to be mitigated by the use of PTA following rather than preceding stent deployment in case reports [50] and multivariate studies, although this may further hinder EPD deployment and the technique was not associated with decreased major adverse event and AIS [27].

The presence of FFT is particularly ominous as the thrombus itself may acutely occlude the ICA or detach and embolize intracranially with minimal or no manipulation [42]. The treatment of these lesions remains controversial, case reports and series have described successful strategies via direct aspiration or mechanical EVT under careful EPD deployment or flow reversal along with delicate CAS placement where systemic therapeutic anticoagulation was determined unreasonable [42,46,48,49,51,52,55,56].

### 3.6. Techniques for Hostile Anatomy

The predominant feared risk in CAS is AIS or TIA [62]. Available literature by way of retrospective case series and reports describes a variety of techniques to safely account for anatomy including unfavorable great vessel variants and angulations [28,29,30,31,32,33,34,35,36,38,39,41], near-total ICA stenosis [33,36,40], tortuosity [28,34,35,37,39], and vulnerable plaque morphology [38] in the context of carotid intervention. Modalities range from anchoring techniques [28], the use of dedicated guide catheters [55], and alternative access sites [30]. Advances in stent design have allowed for flexible systems with combined closed and open cell features and improvements in CEUS and angioscopy allow for timely recognition and management of impending thrombogenic events [91,92]. In patients in which proximal endovascular CCA access is unobtainable, several centers report series using open transcervical approaches versus percutaneous access with an extravascular arteriotomy closure device for direct CCA puncture [31,39] with very low rates of morbidity and mortality on short-term follow-up [39].

Alternative arterial access may improve relative vascular geometry and alleviate some challenges elicited by difficult anatomy. Active investigation continues regarding the use of TFA versus TRA in multiple areas of neuroendovascular surgery [95]. TRA may provide a solution to difficult angulation and arch patterns and avoid arch lesions without the need for direct CCA access with equivocal safety profiles [95]. Analysis was conducted on a general population regardless of baseline anatomy, though it can be assumed that in at least some participating centers TRA was reserved for cases where TFA was thought inadvisable due to hostile vasculature [95]. In multicenter pooled analysis investigating the deployment of intracranial flow diverting stents in the treatment of aneurysms, TRA was found to be associated with significant decreases in access complications and overall complications as compared to TFA [95].

TRA is associated with decreased strain on catheter systems and facilitates navigation of tortuous vessels and effective treatment and EPD system placement in stenotic or vulnerable lesions in appropriate patients [32,34,38]. Reviewed case series showed a rate of stroke of 2.0% across 495 patients in TFA [28,29,31,33,35,36,37,40] and 1.6% across 443 patients in TRA or TBA [30,32,34,38,41]. Univariate comparison of proportions found no significant differences in associated AIS (*p* = 0.6142), suggesting TRA and TBA routes to be viable considerations. Likewise, total complications rates were increased in TFA versus TRA or TBA without statistical significance (4.2% vs 2.7%, *p* = 0.2044). Further study is needed.

### 3.7. Anatomic Guidelines for Endovascular Extracranial Carotid Intervention

Multiple aspects of regional vascular anatomy have been consistently investigated in the context of carotid intervention across heterogeneous data sets [13]. The majority of such characteristics share overlapping associations with one another, advanced age, vascular fragility, environmental factors, and multi-systemic conditions such as diabetes mellitus, hyperlipidemia, hypertension, and others, all of which further contribute to perioperative morbidity and mortality for any intervention. Hostile vasculature appears to have a direct impact on safe intervention in two major contexts, increased procedure length secondary to difficult intra and transluminal access, and safe device deployment secondary to local anatomy. All such factors likely contribute to increased CMT, previously mentioned to be an independent predictor of AIS and MI at one month [17].

In our opinion, CAS under EPD with or without PTA in the hands of the experienced practitioner may serve as an effective primary treatment option for all aspects of ischemic occlusive pathology outside of chronic or near total occlusion. Particular care must be taken in older populations in which prevalence of high-risk factors is increased. Appropriate patient selection and procedural planning has been demonstrated to allow for effective and safe treatment may be achieved in high-risk populations [78].

We recommend consideration of CAS as a primary treatment option for indicated patients on a case-by-case basis dependent on goals of care, acceptable risk profiles as determined by the referring and treating physicians, and patient or surrogate, as well as the familiarity of practicing neuroendovascular teams in alternative access techniques and available devices in individual cases involving high-risk anatomy. Non-emergent cases may benefit from modern intravascular plaque imaging or non-invasive MRI vessel wall sequences to determine degree of risk where available. Patients being considered for recanalization procedures should accordingly be evaluated carefully for CEA when applicable or the use of alternative access sites including the TRA and TBA corridors as well as direct open or percutaneous CCA puncture as appropriate.

Patients with vulnerable plaques (concentric calcification, heterogeneous soft plaque, arch or CCA plaques), severe ICA tortuosity, FFT and near total occlusion should be approached cautiously and with multidisciplinary agreement. We recommend against intervention in chronic total occlusion.

### 3.8. Limitations

We describe a comprehensive literature review without meta-analysis according to PRISMA guidelines. The decision against quantitative meta-analysis was due to highly heterogeneous and overlapping pooled data. In the context of novel techniques or controversial pathology such as FFT, most literature is limited to case reports, which tend to highlight successful procedures in difficult scenarios, making complication and rates outcomes deceivably low in grouped analysis.

## 4. Methods

We queried The United States National Library of Medicine at the National Institutes of Health PubMed database and common internet search engines (Google) for studies involving anatomical considerations in extracranial carotid intervention for non-traumatic non-aneurysmal disease. We found 739 results using the MeSH keywords ‘carotid artery disease’, ‘endovascular procedures’, and ‘mechanical thrombectomy’ on May 26th, 2020. Studies were included if they existed in an English language non-print format and presented or discussed anatomic data in the context of therapeutic endovascular procedures of the CCA, CCA bifurcation, or cervical ICA. Studies were excluded if (1) they discussed extracranial anatomy in the context of intervention for intracranial disease, (2) grouped pathologies were presented with insufficient data provided to separate cases, (3) they discussed traumatic lesions or extracranial carotid aneurysms, (4) they presented data from experimental non-clinical animal, synthetic, or computational models, or (5) they were simply discussions or overviews of methods/techniques with minimal inclusion of objective experimental or clinical data. Abstracts were analyzed qualitatively and independently by authors for review.

## 5. Conclusions

Active debate in patient selection for effective and safe extracranial endovascular carotid intervention for non-traumatic non-vascular disease centers on periprocedural AIS or TIA and unfavorable endovascular anatomy. High-risk characteristics include CCA and arch plaques, heterogeneous soft plaques, severe ICA tortuosity, and concentric calcifications and likely increase risk of ischemic event secondarily by increased CMT. We recommend the selection of endovascular techniques as a primary treatment on an individual basis in cases where available systems and surgeon familiarity provide an acceptable likelihood of rapid access and device deployment following extended discussion with the patient and family when possible. We believe difficult anatomy may be accounted for and overcome prior to intervention in select cases. Further studies using predictive techniques with large-scale data learning sets will help clarify this potential.

## Figures and Tables

**Table 1 jcm-09-03460-t001:** Studies reporting effects of anatomic variables on outcomes in patients undergoing endovascular treatment of extracranial carotid disease.

Study	Population	Tx	Anatomic Criteria	Endpoint	Findings
Dangas et al., 2001 [9] (*n* = 37)	Sx CS	CAS	High CCA bifurcation Contralateral ICA occlusion Previous ipsilateral CEA	AIS (6 months)	No association on univariate analysis.
Lam et al., 2007 [10] (*n* = 133)	Sx CS	CAS PTA	Type III arch Type II arch CCA stenosis CCA tortuosity ICA tortuosity Plaque calcification	AIS, MI or death (1 month)	No association on univariate analysis.
Naggara et al., 2011 [11] (*n* = 262)	EVA-3S	CAS PTA	Type III arch Arch calcification ICA angulation ICA to CCA angulation Ostium involvement Plaque morphology	AIS, MI or death (1 month)	Increased risk of AIS or death associated with ICA to CCA angulation on relative risk ratios.
Werner et al., 2012 [12] (*n* = 751)	Asx/Sx CS	CAS PTA	Type III arch Bovine arch CCA tortuosity ICA tortuosity ICA angulation Concentric calcification	AIS or TIA (NA, while in hospital)	Increased risk of AIS and TIA associated with bovine arch, CCA tortuosity, and ICA tortuosity and angulation on univariate analysis.
Morgan et al., 2014 [13] (*n* = 375)	Sx CS	CAS	Degree of ICA stenosis Arch type Arch calcification Ostium involvement Plaque calcification	AIS or TIA (1 month)	Increased risk of AIS and TIA associated with degree of ICA stenosis on univariate analysis.
Ikeda et al., 2014 [14] (*n* = 50)	Sx CS	CAS	Type III arch Severe CCA tortuosity Severe ICA tortuosity Contra ICA occlusion	New restricting lesions on DWI (24 h)	Increased risk of new restricting lesions on DWI associated with severe ICA tortuosity on multivariate analysis
Fanous et al., 2015 [15] (*n* = 221)	Sx CS	CAS	Arch type Arch calcification ICA calcification ICA tortuosity Ostium involvement Plaque calcification	AIS, MI, or death (1 month)	Increased risk of AIS associated with type III arch. Increased risk of all endpoints associated with ICA tortuosity and concentric calcification of ICA on univariate analysis.
Pelz et al., 2015 [16] (*n* = 181)	Sx CS	CAS	Plaque calcification grade Plaque calcification thickness Proportion of soft plaque	vICA< 120 m/s or vICA:vCCA < 1.4 (30 months)	Increased likelihood of primary outcomes associated with low calcification grade and less thickness and moderate soft plaque on multivariate analysis.
Burzotta et al., 2015 [17] (*n* = 282)	Asx/Sx CS	CAS	Arch type	Catheter time, AIS, or MI (1 month)	Presence of type II or III elongation or bovine arch variants associated with increased catheter manipulation time, which is in turn associated with increased risk of AIS or MI on multivariate analysis.
Doig et al., 2016 [18] (*n* = 115)	Sx CS > 50% (ICSS)	CAS	CCA atheroma ECA atheroma ICA angulation Plaque length Plaque morphology	New restricting lesions on DWI (24 h)	No association on multivariate analysis.
Szikra et al., 2016 [19] (*n* = 101)	Sx CS	CAS	Aortic arch plaque CCA plaque	New restricting lesions on DWI (24 h)	Increased risk of new restriction on DWI associated with arch and CCA plaque presence on multivariate analysis.
AbuRahma et al., 2017 [20] (*n* = 406)	Sx CS	CAS PTA	ICA plaque calcification Type III arch	AIS (1 month)	Increased risk of AIS associated with heavy plaque calcification on multivariate analysis.
Cotter et al., 2019 [21] (*n* = 267)	Asx/Sx CS	CAS	Contralateral ICA occlusion	Restenosis or revascularization after restenosis (5 years)	Increased likelihood of revascularization following restenosis associated with contralateral occlusion.
De Waard et al., 2019 [22] (*n* = 275)	Asx/Sx CS (ICSS)	CAS	Delphi anatomic risk score	AIS (1 month)	No association on multivariate analysis.
Shen et al., 2019 [23] (*n* = 224)	Asx/Sx CS	CAS	Type III arch Type I or II arch	Major AIS (1 month) Minor AIS (1 month) MI (1 month) Death (1 month) CIN (72 h)	Type III arch associated with increased proportion of CIN as compared to combined population of Type I and II arches on univariate analysis.
Zhang et al., 2019 [24] (*n* = 210)	Sx CS	CAS	Degree of unilateral ICA stenosis	Hyperperfusion induced ICH (1 month)	Near total occlusion associated with increased risk of endpoint on univariate analysis.
Hagiwara et al., 2019 [25] (*n* = 11)	Asx/Sx CS	CAS	Plaque enhancement on CEUS Plaque morphology	In-stent intimal hyperplasia (6 months)	Increased risk of hyperplasia with unstable or enhancing plaque morphology on univariate analysis.
Onal et al., 2020 [26] (*n* = 62)	Sx CS.	CAS PTA	Ostium involvement Plaque morphology Pre and post operative ICA angle Change in ICA angle	HD (SBP < 90mmHg or HR < 60 bpm)	Increased risk of HD associated with increased change in ICA angle on multivariate analysis.
Lauricella et al., 2020 [27] (*n* = 309)	Sx CS	CAS PTA	Plaque morphology	Echogenic intraluminal debris (immediate postoperative)	Increased odds of macroscopic debris associated with heterogeneous mainly echoluscent plaque morphology on multivariate analysis.

Tx denotes treatment, Asx and Sx CS asymptomatic and symptomatic carotid stenosis respectively, vICA:ICA mean velocity on ultrasound, ICSS denotes patient data extracted from the international carotid stenting study [7], CEUS carotid enhanced ultrasonography, and HD Hemodynamic depression.

**Table 2 jcm-09-03460-t002:** Reports of novel techniques or modifications in the context of difficult extracranial carotid anatomy in extracranial intervention.

Study	Anatomy	Tx	Technique	Access	Complications
Cardaioli et al., 2009 [28] (*n* = 30)	Type III arch Type II arch Severe tortuosity	CAS/EPD	-Multi-guide wire	CFA	0%
Chang et al., 2009 [29] (*n* = 10)	Type III arch Type II arch	CAS	-DGC	CFA	Total (72 h) 20% AIS (72 h) 10% RD (72 h) 10%
Montorsi et al., 2009 [30] (*n* = 14)	Type III arch Type II arch BC origin of left CCA Common BC-CCA origin	CAS/EPD	-TBA	BA	Retinal embolism (24 h) 8% AIS (1 month) 0% MI (1 month) 0% Death (1 month) 0%
Soloman et al., 2010 [31] (*n* = 12)	Type III arch Type II arch Common BC-CCA origin	CAS/EPD	-DGC	CFA CCA *	0%
Dahm et al., 2010 [32] (*n* = 17)	Type III arch Common BC-CCA origin	CAS/EPD	-TRA	RA	0%
Gonzalez et al., 2011 [33] (*n* = 116)	Near total ICA occlusion	PTA/CAS/EPD	-Standard	CFA	AIS (19 months) 2.6%
Etxegoien et al., 2012 [34] (*n* = 347)	Type III arch Type II arch BC origin of left CCA Common BC-CCA origin Severe tortuosity	CAS/EPD	-TRA	RA	AIS (1 month) 1.6%
Barbiero et al., 2013 [35] (*n* = 37)	Type III arch Type II arch Bicarotid trunk and aberrant subclavian BC origin of left CCA Common BC-CCA origin Bovine arch Severe tortuosity	PTA/CAS/EPD	-DGC	CFA	AIS (24 h) 5.4% AIS (1 month) 0% MI (1 month) 0%
Sakamoto et al., 2013 [36] (*n* = 14)	Near total ICA occlusion	CAP/CAS/EPD	-PBO	CFA	DWI restriction (33 months) 14.3% AIS (33 months) 0%
Hopf-Jensen et al., 2014 [37] (*n* = 94)	Severe tortuosity Contra ICA occlusion	CAS/PTA	-Stent protected PTA	CFA	Total (1 month) 2.9% Death (1 month) 1.0% AIS (34 months) 0% Stent stenosis (34 months) 2.1%
Montorsi et al., 2014 [38] (*n* = 60)	Type III arch Type II arch BC origin of left CCA Common BC-CCA origin Severe soft plaque morphology	CAS/EPD	-TBA/TRA	BA RA	Retinal embolism (24 h) 1.1% AIS (1 month) 0% MI (1 month) 0% Death (1 month) 0% Total (18 months) 7% Vascular injury (18 months) 2.2%
Bergeron et al., 2015 [39] (*n* = 306)	Type III arch Type II arch BC origin of left CCA Common BC-CCA origin Bovine arch Severe tortuosity	PTA/CAS + -EPD	-Open TCA -Percutaneous TCA	CCA CCA *	AIS (1 month) 0.6% (open only) Death (1 month) 0% MI (1 month) 0%
Akkan et al., 2018 [40] (*n* = 182)	Near total ICA occlusion	PTA/CAS/EPD	-Standard	CFA	AIS (1 month) 2.2% MI (1 month) 0% Death (1 month) 0% Stent stenosis (64 months) 3.8%
Koge et al., 2018 [41] (*n* = 5)	Type III arch Type II arch BC origin of left CCA Common BC-CCA origin Acute takeoff of CCA	PTA/CAS/EPD	-TBA -PBO	BA	Death (1 month) 20% Pseudoaneurysm (1 month) 20%

BC-CCA denotes a common origin of the brachiocephalic and left common carotid arteries, RD respiratory depression, BC brachiocephalic artery, BA brachial artery, TBA transbrachial access, TRA transradial access, TCA transCCA access, * CCA access by open transcervical cutdown, PBO proximal balloon occlusion, and DGC dedicated guide catheter.

**Table 3 jcm-09-03460-t003:** Case reports regarding extracranial carotid intervention and unusual anatomy or clot morphology.

Study	Age/Sex	Anatomy	Pathology	Tx	Access	Outcome
Parodi et al., 2005 [42]	51/F	FFT	Sx ICA stenosis (80%)	-EVT -PTA/CAS under FR	CFA	Resolution
Gupta et al., 2008 [43]	NA/NA	Type III arch Ulcerated plaque	Sx ICA stenosis (80%)	-CAS under EPD	CFA	NA
Kassaian et al., 2008 [44]	76/M	Type III arch Ulcerated plaque	Sx ICA stenosis (70%)	-CAS under EPD	CFA	Resolution
Gan et al., 2010 [45]	78/F	BC-CCA	Sx ICA stenosis (90%)	-CAS under EPD	RA	Resolution
Park et al., 2011 [46]	55/F	FFT	Sx distal CCA stenosis (50%)	-CAS under EPD	CFA	Resolution
	75/M	FFT	Sx mid CCA stenosis (50%)	-CAS under EPD	CFA	Resolution
	55/F	FFT	Sx CCA bifurcation stenosis	-EVT under PBO and EPD	CFA	Resolution
Barutcu et al., 2013 [47]	64/F	Bovine arch ^a^	Symptomatic ICA stenosis (95%)	-PTA/CAS -ECA anchoring	CFA	Resolution
Tan et al., 2014 [48]	44/M	FFT	Sx CCA bifurcation stenosis (20%)	-EVT under EPD	CFA	Resolution
Giragani et al., 2017 [49]	45/M	FFT	Sx Mid CCA stenosis	-EVT under EPD	CFA	Persistent hemiparesis
Huntress et al., 2017 [50]	72/M	Type III arch	Sx ICA stenosis (80%)	-CAS under EPD with post-stent PTA	CCA *	Improving hemiparesis
Carr et al., 2018 [51]	51/F	FFT	Sx Mid CCA stenosis (85%)	-EVT	CFA	Improved
Bae et al., 2018 [52]	72/M	Mobile intimal flap	ASx ICA stenosis (80%)	-PTA/CAS under EPD	CFA	Resolution
	75/M	Mobile intimal flap Irregular calcified plaque	ASx ICA stenosis (80%)	-PTA/CAS under EPD	CFA	No new deficit
Kim et al., 2019 [53]	70/M	ICA APA origin	Sx ICA Stenosis (99%)	-PTA/CAS under EPD	CFA	Improving aphasia
Lin et al., 2019 [54]	45/M	Thoracic Aortic Dissection, Type A	Acute ICA LVO	-Multiple telescoping CAS and EVT	CCA *	Resolved
Morr et al., 2019 [55]	70/M	FFT Tortuosity	Sx Severe ICA stenosis	-CAS under FR and ECA PBO	CCA *	No new deficit
	79/M	Tortuosity Intraluminal thrombus	Sx Severe ICA stenosis	-PTA/CAS under FR and ECA PBO	CCA *	No new deficit
Yamaoka et al., 2019 [56]	77/F	FFT	Sx CCA bifurcation stenosis	-EVT under proximal CCA and distal ICA PBO	CFA	No new deficit

NA denotes information not provided, PBO proximal balloon occlusion and ^a^ single trunk from transverse thoracic aorta with distal branching into bilateral subclavian and common carotid arteries. *CCA access by open transcervical cutdown, PBO proximal balloon occlusion, and DGC dedicated guide catheter.

**Table 4 jcm-09-03460-t004:** Reports describing changes in anatomy following extracranial carotid intervention.

Study	Tx	Anatomical Changes	Consequences/Conclusions
Vos et al., 2005 [57] (*n* = 6)	CAS	Forward head position associated angulation of ICA (+10.2 degrees)	-No clinical consequences on univariate analysis
Shakur et al., 2015 [58] (*n* = 18)	CAS	Increased vICA with no significant change in stenosis or vMCA	-Restoration of vICA with CAS on univariate analysis
Ohshima et al., 2017 [59] (*n* = 1)	CAS	Acute progression of ulcerated plaque to stent thrombosis	-Atheroembolic AIS reported
Chen et al., 2018 [60] (*n* = 17)	CAS	Increased CBF and CVR in patients with preoperative impairments	-Hemodynamic benefits dependent on severity of disease on univariate analysis
Tanaka et al., 2018 [61] (*n* = 11)	CAS	Velocities of intracranial vessels increase at one month following CAS and decrease to preoperative levels at 3 months	-CAS results in temporary increased flow rates in large intracranial vessels

MCA denotes middle cerebral artery, vICA ICA flow velocity, CBF cerebral blood flow, CVR cerebrovascular reactivity.

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
