# Peer review of "Anatomical Considerations for Endovascular Intervention for Extracranial Carotid Disease: A Review of the Literature and Recommended Guidelines"

_jcm, 2020, doi:10.3390/jcm9113460_

Round 1

Reviewer 1 Report

The article provides a succinct summation of details regarding a multitude of factors affecting potential complications associated with endovascular intervention for extracranial carotid disease.While it does not include any novel points or conclusions, it provides readers with a brief outline of the current literature, which may be informative. Tables 1 and 2 effectively outline the general findings of each included study regarding associations with particular anatomic variations, plaque morphology, etc, however, further statistical analysis or meta-analysis of the pooled data would provide the reader with more applicable data and contribute to the discussion. Moreover, further analysis of case reports of individual patients/complications listed in Table 3 should be completed to allow readers to extrapolate pertinent information. These revisions would allow the article to be less anecdotal with an evidence-based perspective.

Reviewer 2 Report

The authors give a comprehensive overview over the litrature regarding endovascular intervention for extracranial carotid disease.

The authors give a comprehensive overview over the literature regarding endovascular intervention for extracranial carotid disease. They should be complimented for their work. According to a short search by myself they did not miss highly relevant sources.

In my opinion this overview adds some insight to the topic. A meta-analysis would indeed be problematic due to the heterogeneity of the data. However, they could have accounted for this and performed meta-analysis in subgroups of the papers.

Their discussion and conclusions seem valid.

Reviewer 3 Report

The paper, particularly the discussion appears a bit lenghthy.

The study addressed the question whether particular anatomical constellations are associated with  catheter  stenting of the extracranial carotid artery, The analysis was done as a narrative review of the published literature.  A more quantitative meta-analysis was not performed

The topic is of technical nature. One would expect it therefore in a more specialized  (neuroradiological) journal. The results are not really surprising, more of confirmatory nature for the specialist in the field.

The topic is of moderate originality.

It summarizes experiences of institutional series.

The paper is well written.  The Discussion is somewhat lengthy.

The text is clear and easy to read

The conclusions are consistent with the evidence and arguments presented

The authors address the main question posed

Round 2

Reviewer 1 Report

The article fails to significantly contribute to the field as it is overly comprehensive without practice-changing recommendations or conclusions.

Author Response

We appreciate the reviewer's comment. We respectfully disagree with the opinion concerning significance however, we provide a comprehensive review via PRISMA standards on a topic under intense active debate vascular and cerebrovascular literature and offer recommendations based on available sources. We have edited down redundant language and addressed prior concerns. We would welcome further specific constructive feedback as to how we can improve our report.